

# Characterization of atmospheric bioaerosols along the transport pathway of Asian dust during the Dust-Bioaerosol 2016 Campaign

Kai Tang[1], Zhongwei Huang[1*], Jianping Huang[1], Teruya Maki[2], Shuang Zhang[1], Xiaojun Ma[1], Jinsen Shi[1], Jianrong Bi[1], Tian Zhou[1], Guoyin Wang[1], Lei Zhang[1]

[1]Key Laboratory for Semi-Arid Climate Change of the Ministry of Education, College of Atmospheric Sciences, Lanzhou University, Lanzhou, 730000, China.
[2]College of Science and Engineering, Kanazawa University, Kakuma, 920-1192, Japan

*Correspondence to*: Zhongwei Huang (huangzhongwei@lzu.edu.cn)

## Abstract

Previous studies have shown that bioaerosols are injected into the atmosphere during dust events. These bioaerosols may affect leeward ecosystems, human health and agricultural productivity and may even induce climate change. However, bioaerosol dynamics have rarely been investigated along the transport pathway of Asian dust, especially in China, where dust events affect huge areas and massive numbers of people. Given this situation, the Dust-Bioaerosol (DuBi) Campaign was carried out over

northern China, and the effects of dust events on the amount and diversity of bioaerosols were investigated. The results indicate that the number of bacteria showed remarkable increases during the dust events, and the diversity of the bacterial communities also increased significantly, as determined by means of microscopic observations with 4,6-diamidino-2-phenylindole (DAPI) staining and MiSeq sequencing analysis. These results indicate that dust clouds can carry many bacteria of various types

into downwind regions and may have potentially important impacts on ecological environments and climate change. The abundances of DAPI-stained bacteria in the dust samples were one to two orders of magnitude greater than those in the non-dust samples and reached $10^5 \sim 10^6$ particles·m$^{-3}$. Moreover, the charge capacity of yellow fluorescent particles associated with the DAPI-stained bacteria increased

from 5.1% ± 6.3% (non-dust samples) to 9.8% ± 6.3% (dust samples). A beta diversity analysis of the

bacterial communities demonstrated the distinct clustering of separate prokaryotic communities in the

dust and non-dust samples. *Actinobacteria*, *Bacteroidetes*, *Proteobacteria* remained the dominant phyla

in all samples. As for Erenhot, the relative amounts of *Acidobacteria* and *Chloroflexi* have a remarkable

rise in dust events. *Alphaproteobacteria* made the major contribution of the increasing relative amounts

of the phylum *proteobacteria* in all dust samples. In the future, the viability and activity of airborne

microbes, the interactions between bioaerosols and other gaseous and solid components in the air, and

the effects of bioaerosols on animals and plants, ecological environments and the climate system must

be studied in depth to help us understand the behavior of bioaerosols in the air and dust clouds in greater

detail.

## 1 Introduction

Bioaerosols are a class of atmospheric particles that range in size from nanometers up to about a

tenth of a millimeter. They are made up of living and dead organisms (e.g., algae, archaea, and bacteria),

dispersal units (e.g., fungal spores and plant pollen), and various fragments or excretions (e.g., plant

debris and brochosomes) (Fröhlich-Nowoisky et al., 2016). Several studies have investigated the role of

dust events as a vehicle for bioaerosols (Hervàs et al., 2009; Prospero et al., 2005; Yamaguchi et al.,

2012). Asian dust events are capable of moving masses of soil-derived dust over long distances and may

introduce large amounts of microorganisms and pollen to the atmosphere. It is well known that Asian

dust frequently disperses all around the East Asian regions (Iwasaka et al., 1983) and can even reach the

Americas (Husar et al., 2001) and the Arctic (Huang et al., 2015). Asian dust clouds can sometimes be



transported more than one full circuit around the globe in approximately 13 days (Uno et al., 2009), and Asian dust has been identified in ice and snow cores from Greenland (Bory et al., 2003) and the French Alps (Grousset et al., 2003). Climate change (Prospero and Lamb, 2003) and anthropogenic causes (Neff et al., 2008) may increase the magnitude and frequency of dust storms in semi-arid and arid

regions in the near future. Increasing evidence shows that microbes are transported by Asian dust events (Hua et al., 2007; Maki et al., 2017; Yeo and Kim, 2002). In Japan, the concentrations of bacterial cells and the structure of airborne bacterial communities in the near-surface air and the free troposphere are affected by Asian dust events (Maki et al., 2014, 2015). Similarly, the yellow sandstorms that originate in Asian deserts have been reported to affect the ambient air quality of Taiwan by increasing the levels

of fungal spores (Ho et al., 2005; Wu et al., 2004). Results from South Korea also show that Asian dust impacts both airborne fungal concentrations and fungal communities (Jeon et al., 2011, 2013; Yeo and Kim, 2002).

Considering the transoceanic and transcontinental dispersal of bioaerosols associated with dust events, the importance of bioaerosols in the atmosphere is likely to be seriously underestimated. These

bioaerosols may affect leeward ecosystems, human health, and agricultural productivity, and they may play a larger role in the climate system by acting as efficient ice nucleating particles and cloud condensation nuclei. Certain species of bacteria and fungi are known to have very high ice nucleating (IN) ability, especially at warmer temperatures (Maki and Willoughby, 1978), potentially leading to the initiation of ice formation in clouds and thereby influencing precipitation, cloud dynamics and the

amount of incoming and outgoing solar radiation (Creamean et al., 2013). A study used a cloud simulation chamber to demonstrate that bacterial IN activity is maintained even after cell death (Amato

et al., 2015). Hence, the role of microorganisms in the atmosphere is an underappreciated aspect of biological and atmospheric science; these microorganisms have potentially important impacts on the hydrological cycle, clouds, and climate (DeLeon-Rodriguez et al., 2013).

Moreover, bioaerosols may have a significant influence on human health and the spread of plant diseases. Airborne microorganisms containing bacteria, fungi and viruses can have infectious, allergenic, or toxic effects on living organisms, causing disease or allergies in humans, agricultural crops, livestock, and ecosystems, including coral reefs. The dust event-driven dispersal of bioaerosols is strongly correlated with allergen burdens and asthma (Griffin, 2007; Ichinose et al., 2005; Liu et al., 2014). The long-distance aerial dispersal of pathogens by the wind can spread plant diseases (Brown and Hovmøller, 2002) and human diseases, such as Kawasaki disease (Jorquera et al., 2015; Rodo et al., 2011).

Therefore, information on the abundance of bioaerosols in this dust is necessary to assess the influence of these bioaerosols on public health, ecosystems, biogeographical distributions, and meteorological and climatic processes (Hara and Zhang, 2012). Recent research has showed that the Gobi Desert of Asia, instead of the Taklimakan Desert, plays the most important role in contributing to dust concentrations in East Asia, and approximately 35% of the dust emitted from the Gobi Desert of Asia is transported to remote areas in East Asia in spring (Chen et al., 2017). To investigate the effects of dust events from the Gobi Desert of Asia on the amount and diversity of microbes in the air, the Dust-Bioaerosols (DuBi) Campaign was carried out during March through May in 2016. This campaign is named "DuBi-2016" in this paper (Fig. 1).

In the DuBi-2016 campaign, air sampling was performed continuously at three sites downwind of the Gobi Desert of Asia. These sites lie along the transport path of Asian dust, which were Erenhot, Zhangbei and Jinan. Frequent dust storms attacked Erenhot directly, and a great amount of transported dusts was observed there. Only a small number of them, by contrast, could arrive in Zhangbei and Jinan.

5 In addition, some samples were collected on the road between Dalanzadgad and Ulaanbaatar, and these samples enable comparison of the structure of microbial communities between the source and downwind regions. Through combining microscopy and MiSeq sequencing analysis, the potential effects of long-range transported dust on the amount and diversity of bioaerosols can be well characterized.

## 2 Experiments

### 2.1 Sample collection

Information on the sampling sites is provided in Table 1. The sampling sites in Erenhot and Zhangbei are located to the northwest of the residential area and at a distance from this area. Anthropogenic activities that might influence the sites are not expected in cases in which air masses

15 arrive from the south, southwest, west, or northwest. Therefore, the dust particles appearing at the sites had traveled long distances in the atmosphere and originated primarily in Mongolia and northern China. In addition, five bioaerosol samples were collected on the car along the road between Dalanzadgad and Ulaanbaatar (R-DzToUb). These samples represent conditions in the dust source regions.

The bioaerosol samples were collected using four sterilized polycarbonate filters with a pore size

20 of 0.2 µm (Whatman 111106, China) with a sterilized Swinnex 13-mm filter holder (Millipore

SX0001300, China) connected to an air pump (AS ONE MAS-1, Japan; the flow rate for each filter was approximately 0.3 L·min$^{-1}$) for 1~24 h, according to air quality conditions. Whenever dust arrived, intensive observations were made to get the information on the fine structure of the dust event. Before sampling, all of the filters were sterilized by autoclaving (121°C for 20 min). After sampling, the samples were stored at -80 °C until the downstream analyses were performed.

To avoid contamination, the sampling filter holder and the materials used to change the filters were treated with 75% ethanol every day, and a mask was worn during operation. Detailed information on the samples is provided in Table S1, Table S2, and Table S3.

## 2.2 Meteorological data and aerosol information

In Erenhot, a TR-74Ui device (T&D Corporation, Japan) was used to measure the temperature, relative humidity, illuminance and UV intensity sequentially. Data describing the attenuated backscatter coefficient, the volume depolarization ratio, and the color ratio were obtained from the Zamynuud observation site of AD-Net (43.72° N, 111.90° E, 962 m ASL), which is located less than 10 km away from the sampling site in Erenhot (Nishizawa et al., 2016).

In Zhangbei, basic meteorological information, including measurements of temperature, relative humidity, pressure, wind, precipitation and radiation, was gathered by an automatic meteorological station (weather transmitter WXT520, Vaisala), and the PM$_{2.5}$ mass concentrations were measured using a continuous ambient particulate TEOM™ Monitor (Series 1400a, Thermo Fisher Scientific Inc.) (Bi et al., 2017; Jianping and Xin, 2008; Wang et al., 2010).



Seventy-two-hour backward trajectories of the air masses at the Zhangbei observational site were calculated using the National Oceanic and Atmospheric Administration Hybrid Single Particle Lagrangian Integrated Trajectory (HYSPLIT) model (http://www.arl.noaa.gov/HYSPLIT.php).

### 2.3 Sample analysis

The total number of microorganisms in the bioaerosols was determined by a modified counting method that was previously described by Maki (Maki et al., 2014). The samples were stained with 10 μg·mL$^{-1}$ 4,6-diamidino-2-phenylindole (DAPI; D9542, Sigma, China; the DAPI-DNA complex has an excitation wavelength of 364 nm and an emission wavelength of 454 nm) for 15 min after being fixed in a 4% paraformaldehyde solution for 1 h. The filter was then placed on a slide in a drop of low-

fluorescence immersion oil (IMMOIL-F30CC, Olympus). A second drop of oil was added, and a coverslip was placed on top. Next, the prepared slides were observed using an epifluorescence microscope (BX53 and DP72, Olympus, Japan) equipped with an ultraviolet excitation system; an excitation waveband of 340~390 nm was used. Fluorescent particles with four different colors, blue, white, yellow, and black, were counted in 10 randomly selected fields. The fluorescent particle

concentrations in the bioaerosols were calculated using the following formula.

$$C = \frac{S_1 \times N_0}{S_0 \times V},$$

(1)

where $C$ is the number of fluorescent particles in the bioaerosols (particles·m$^{-3}$), $S_1$ is filtration area on the membrane ($7 \times 10^7$ μm$^2$), $S_0$ is the area of each microscopic field ($1.46 \times 10^4$ μm$^2$), $N_0$ is the average





number of fluorescent particles in the microscopic field, and $V$ is the volume of the filtered sample (m³).

The detection limit of the particles is approximately $1.0 \times 10^4$ particles m⁻³ of air.

## 2.4 DNA extraction, sequencing and phylogenetic analysis

The genomic DNA (gDNA) was extracted from the atmospheric samples from Erenhot and
Mongolia using the PC extraction/alcohol precipitation method. Two-step PCR amplification and
product purification were then carried out according to the method of Maki (Maki et al., 2017). Two-
step PCR has several advantages, such as increased reproducibility and the recovery of greater levels of
genetic diversity during amplicon sequencing (Park et al., 2016). An Illumina MiSeq sequencing system
(Illumina, CA, USA) and a MiSeq Reagent Kit V2 (Illumina, CA, USA) were used to perform the
sequencing, and an average read length of 270 bp was obtained. All the data obtained from MiSeq
sequencing have been deposited in the DDBJ/EMBL/GenBank database, and the accession number of
the submission is PRJNA413598.

The R software package (version 3.4.1) was employed to analyze the experimental data. The
"phyloseq" package (version 1.20.0) was used to handle and analyze the high-throughput sequencing
data. The Shannon index (H′) and the Simpson index (D) are calculated as follows:

$$H' = -\sum_{i=1}^{S} P_i * log_2 P_i,$$

(2)

$$D = 1 - \sum_{i=1}^{S} (P_i)^2,$$

(3)

where $S$ is the number of operational taxonomic units (OTUs), and $P_i$ is the relative proportion of an
individual species $i$.

Principal coordinate analysis (PCoA) with weighted UniFrac distances was used to explore and visualize similarities or dissimilarities of the bacterial communities contained in samples. UniFrac measures the difference between two collections of sequences as the amount of evolutionary history that is unique to either of the two, which is measured as the fraction of branch length in a phylogenetic tree that leads to descendents of one sample or the other but not both (Lozupone et al., 2011). There are two phylogenetic measures of community β diversity: unweighted UniFrac, a qualitative measure, which use only the presence/absence of data, and weighted UniFrac, a quantitative measure, which use the abundance of each taxon (Lozupone et al., 2007). UniFrac, coupled with standard multivariate statistical techniques including principal coordinates analysis (PCoA) can be used to cluster many samples according to the difference of their bacterial communities.

## 3 Results and discussion

### 3.1 Identification of the dust events

The dust events in spring 2016 were identified from lidar observations made in Zamynuud and $PM_{2.5}$ mass concentrations measured in Zhangbei. In addition, meteorological factors, such as atmospheric pressure, wind speeds and wind directions, were checked to confirm the dust events in Zhangbei. The attenuated backscattering coefficient at 532 nm (Att. Bac. Coe), the volume depolarization ratio (Dep. Rat) and the color ratio (Col. Rat.) increased dramatically when the dust events occurred. Seven heavy dust events (D1-D7) occurred in Erenhot during the sampling period (Fig. 2). Accordingly, the samples collected during events D1-D7 are named the "dust samples" (Table S1). During events D2, D3 and D7, the mass concentrations of $PM_{2.5}$ in Zhangbei increased significantly

with northwest or north winds, increasing wind speed and an apparent decline of atmospheric pressure

(Fig. 3). These observations indicate that dust events occurred in Zhangbei at that time. A slight

increase in $PM_{2.5}$ mass concentrations was observed during event D6, accompanied by a strong north

wind, indicating that Zhangbei was slightly affected by the dust events that occur in Erenhot.

Accordingly, samples ZB3_31N, ZB4_6D, ZB4_6N, ZB4_21D were considered as "dust samples"

(Table S2). The 72-h back trajectories of air masses in Zhangbei calculated using the HYSPLIT model

indicate that the dust events D2, D3, D6 and D7 originated in the Gobi Desert of Asia and passed over

Erenhot and Zhangbei during the transport process. Several peaks in $PM_{2.5}$ concentrations appeared in

Zhangbei on Mar. 30, Apr. 4 and Apr. 11. These peaks were not dust events, as determined using the

10 wind speeds, wind directions and 72-h back trajectories (Figs. 3 and 4). An air pollution event named

"P1" occurred on Apr. 11 that was characterized by high $PM_{2.5}$ concentrations, strong south winds and

air masses that originated in the southern regions (Fig. 4).

### 3.2 State of the bioaerosols in the dust and non-dust samples

  Under microscopic observation, the particles stained with DAPI emitted several types of

15 fluorescence, mainly blue, white, yellow, or black fluorescence (Fig. 5). These particles were thus

categorized as DAPI-stained bacteria (with diameters < 3 µm), white particles (mineral particles),

yellow particles (organic matter) and black particles (black carbon) (Maki et al., 2017). Analysis of the

microphotographs shows that the dust and non-dust samples were significantly different. As two

examples, ER4_12 was compared with ER4_13, and ER4_15D1 was compared with ER4_16.

Compared with sample ER4_16, which was collected during a non-dust event, dust sample ER4_15D1

contained a surprising number of DAPI-stained bacteria (coccoid- or bacillus-like bacteria) (Fig. 6a and





b). This comparison clearly demonstrates that dust events can carry large amounts of bacteria into the atmosphere, and these microbes continue to float towards downwind regions. We also take the dust events that occurred on Apr. 12 and 13 as another example. These events differed somewhat from each other in terms of their dust intensity and dust blowing height. The lidar data clearly demonstrate that the

dust mass noted on Apr. 12 fell to the ground from nearly 4 km, whereas the dust event that occurred on Apr. 13 was mild by contrast and likely originated from a local source. The sample ER4_12D contained more DAPI-stained bacteria, although the sampling duration was shorter than that of sample ER4_13 (Fig. 6c and d). This result illustrates that dust transported over long distances contains large amounts of microorganisms and may have substantial impacts on downwind regions. In addition, epifluorescence

microscopy has revealed that aerosols collected at 800 m over the Taklimakan Desert contained large particles attached with microorganisms, such as bacteria (Maki et al., 2008). Airborne microbes are often attached to larger particles especially yellow particles and found as agglomerates (Tong and Lighthart, 2000), which may help them survive nutrient shortages and UV radiation and may even facilitate the growth and reproduction of the microbes.

Under the SEM, the bioaerosols were seen to display various states (Fig. 7). Spiny fungal spores (Fig. 7a) and shriveled pollen (Fig. 7d) may represent strategies that organisms take to protect themselves from the harsh atmospheric environment and ensure their survival. Entering a non-dividing state (dormancy) in which they transform morphologically to spores or undergo other cell wall modifications and slow down or stop their metabolic activity can improve their resistance to physical

stresses, such as desiccation and UV radiation, which increases their chances of survival in the atmosphere (Smets et al., 2016). Furthermore, some coccus-like airborne microbes were found attached



to mineral particles (Fig. 7e and f), which may serve as shelters and favor the survival of the bacteria. Interestingly, exocytosis or endocytosis (Fig. 7c) and cell division processes (Fig. 7b) were captured, showing that microbial activity proceeds in the air.

### 3.3 Variations in the concentrations of fluorescent particles in the dust and non-dust samples

When the Asian dust events occurred in downwind area, airborne microbial abundances increased at 10- or 100-folds (Hara and Zhang, 2012), and showed relative correlation with $PM_{10}$, which are indicators of dust occurrences (Dong et al., 2016, Cha et al., 2016 and 2017). In this study, the concentrations of the DAPI-stained bacteria, the white particles and the yellow particles in the dust samples were significantly higher than those in the non-dust samples, whereas the concentrations of the

black particles (black carbon) showed no obvious pattern, regardless of the occurrence of dust events (Fig. 8a, b and c). In general, the concentrations of DAPI-stained bacteria in the non-dust samples were on the order of $10^4$ to $10^5$ particles·m$^{-3}$, whereas those in the dust samples were $10^5$ to $10^6$ particles·m$^{-3}$. These concentrations are similar to the results of other field observations made in Tsogt-Ovoo in the Gobi Desert of Asia (Maki et al., 2016) and indicate that dust events can carry abundant microbes. In

addition, the concentrations of the yellow particles in the non-dust and dust samples were on the order of $10^5$ to $10^6$ particles·m$^{-3}$ and $10^6$ to $10^7$ particles·m$^{-3}$, respectively. Aerosols transported by Asian dust events are reported to include high amounts of organic molecules, such as mannitol, glucose, and fructose, which consist of the cell components of airborne microorganisms (Fu et al., 2016). Similarly, the concentrations of the yellow particles (organic matter) and the white particles (mineral particles)

increased with the concentrations of DAPI-stained bacteria (Fig. 9a), whereas the concentrations of the black particles showed no obvious pattern (Fig. 9a and b). These observations indicate that a close




relationship exists among DAPI-stained bacteria, organic matter and mineral particles. It is speculated that the yellow particles (organic matter) and the white particles (mineral particles) serve as nutrient sources and shelters for microbes, respectively, and favor their survival and long-distance transport. Furthermore, the concentrations of the black particles decreased in some of the dust events (Fig. 8a),

and the concentrations of the DAPI-stained bacteria and the yellow particles showed no obvious relationship with the concentrations of the black particles, whereas the concentrations of the white particles showed a declining trend (Fig. 9b). Thus, fewer black particles existed in the dust samples, and the black particles have little connection with the DAPI-stained bacteria and the yellow particles. These results are supported by other field observations in Tsogt-Ovoo of the Gobi Desert of Asia (Maki et al.,

2016).

The ratio of the concentrations of the DAPI-stained bacteria, the black particles and the white particles to those of the yellow particles were considered together with the charge capacity of the yellow particles. The charge capacity of the yellow particles for the DAPI-stained bacteria increased slightly with the concentrations of the yellow particles (Fig. 10a). This result indicates that the yellow particles

(organic matter) in the dust may serve as nutrient sources and favor microbial survival, which is also partly confirmed by the micrographs (Fig. 6a, b, c and d). The ratios of the concentrations of the DAPI-stained bacteria, the black particles and the white particles to those of the yellow particles ranged from 5.1% ± 6.3% (non-dust samples) to 9.8% ± 6.3% (dust samples), from 73.6% ± 100.4% (non-dust samples) to 9.0% ± 8.2% (dust samples), and from 2.7% ± 3.3% (non-dust samples) to 3.8% ± 4.1%

(dust samples), respectively (Fig. 10b). It is quite clear that the charge capacity of the yellow particles associated with the DAPI-stained bacteria was higher in the dust samples compared with that in the



non-dust samples. On the other hand, the charge capacity of the yellow particles associated with the black particles was much lower in the dust samples; thus, greater numbers of bacteria can be contained in a unit of yellow particles during dust events, whereas the black particles displayed the opposite behavior.

## 3.4 Alpha and beta diversity analysis of the samples

The 16S rDNA sequences from 22 samples were divided into 28,949 OTUs (sequences with > 97% similarity), and the number of OTUs contained in these samples ranged from 150 to 3147 (Fig. 11a). On the other hand, the ITS rDNA sequences from 18 of the samples were divided into 223 OTUs, and the number of OTUs contained in these samples ranged from 5 to 74 (Fig. 11b). Phylogenetic assignment of the 16S rDNA sequences resulted in an overall diversity profile that included bacteria and archaea, 34 phyla and candidate divisions, 94 classes (and class-level candidate taxa), 166 orders (and class-level candidate taxa), and 243 families (and family-level candidate taxa). Phylogenetic assignment of the ITS rDNA sequences resulted in an overall diversity profile that included 3 phyla (*Ascomycota*, *Basidiomycota*, and *Chytridiomycota*), 19 classes (and class-level candidate taxa), 62 orders (and class-level candidate taxa), and 149 families (and family-level candidate taxa). Overall, the alpha diversity of the bacteria in the dust samples was higher than that of non-dust samples collected in Erenhot (Fig. 12a), whereas the alpha diversity of the fungi was much lower and showed no obvious pattern between the dust and non-dust samples (Fig. 12b). The results from another study in South Korea also suggest that airborne bacterial diversity (at least the richness index) is increased during Asian dust events (Cha et al., 2016). It illustrates that the dust events can carry not only a huge number of bacteria, but also a great variety of that.





To analyze similarities in the bacterial community contained in each sample, principal coordinates analysis (PCoA) with weighted UniFrac distances was carried out. The sample "Dz5_5R100" was discarded before PCoA analysis, due to the small number of OTUs it contained (Fig. 11a). The results indicate that the dust and non-dust samples from Erenhot display a distinct separation (Fig. 13), which

indicates that the bacterial community compositions differed significantly between the dust and non-dust samples. Similarly, the distinct clustering of prokaryotic communities separating dust and non-dust samples of Tsogt-Ovoo was found in another study (Maki et al., 2016). Furthermore, the dust samples collected in Erenhot showed a high degree of similarity with the samples of R-DzToUb, which suggest that some types of bacteria were transported from the Gobi Desert of Mongolia to Erenhot.

**3.5 Analysis of the microbial community composition in the dust events and non-dust events**

Comparative analysis of the bacterial community composition revealed that the ubiquitous bacterial phyla in the air were *Acidobacteria*, *Actinobacteria*, *Bacteroidetes*, *Chloroflexi*, *Firmicutes*, *Gemmatimonadetes*, and *Proteobacteria* (Fig. S4), which are typically the most abundant phyla in the atmospheric environment of the Gobi Desert (Maki et al., 2016). Of these phyla, *Actinobacteria*,

*Bacteroidetes*, *Proteobacteria* remained the dominant phyla in all samples (Fig. 14).

At the phylum level, as for Erenhot, the relative amounts of *Acidobacteria* and *Chloroflexi* in dust samples have a remarkable rise compared with non-dust samples, and the next were *Crenarchaeota*, *Firmicutes* and *Proteobacteria* (Fig. 14a and b). By contrast, opposite phenomenon appeared in R-DzToUb, and the relative amounts of *Acidobacteria*, *Chloroflexi* in non-dust samples were greater than

that in dust samples (Fig. 14c and d). The class *Chloroflexi* was dominating among the members of this phylum (Fig. 14 and 15). It's worth noting that all samples (dust and non-dust samples) of R-DzToUb



were collected on the road from Dalanzadgad to Ulaanbaatar, small dust events continuously occur there and some residues of dust particles would be remained in the air for a longer period. Notably, the phylum *Chloroflexi* includes six classes, of which only the class *Chloroflexi* consists of phototrophic bacteria. This phototrophic group, called filamentous anoxygenic phototrophic bacteria, shares the

5 following features in common: multicellular filamentous morphology, gliding motility, and anoxygenic photosynthetic activity (Hanada, 2014). This phototrophic group could obtain light energy and keep their survival in the air, using the light for photosynthesis. Long-distance transmission of such group is possible. The relative amounts of *Proteobacteria* in all dust samples increased slightly compared with non-dust samples, and among this group *Alphaproteobacteria* made the major contribution, by contrast,

*Gammaproteobacteria* was just the reverse (Fig. 14 and 15). It suggests that the relative amounts of *Alphaproteobacteria* in the dust are higher than that in the air. These bacteria could help to identify the mixture levels of air masses transported for long distances, even the relative contributions of local sources and remote sources (particularly deserts) to the concentration of airborne biological particles in different regions (Maki et al., 2017).

At the class level, *Chloracidobacteria* and *Gemmatimonadetes* in dust samples of Erenhot and non-dust samples of R-DzToUb have higher relative amounts compared with non-dust samples of Erenhot and dust samples of R-DzToUb (Fig. 15). *Cytophagia* in the phylum *Bacteroidetes* shows similar phenomenon. Further, *Bacilli* in non-dust samples of Erenhot shows very low amounts down to the detection limit, whereas its relative amounts in other samples keep stable.

In addition, two phyla in the archaea kingdom, *Crenarchaeota* (which contains *Thaumarchaeota* at the class level) and Euryarchaeota (which contains *Methanobacteria*, *Thermoplasmata*,

*Methanomicrobia* and *Halobacteria*), were detected, but their abundances were relatively low compared to the dominant bacterial phyla. Particularly, *Thaumarchaeota* was found only in the samples "ER3_31N" (heavy dust event) and "Dz5_5R600" (the dust source region) in proportions exceeding 2% (Fig. S4), which implies that it may be used as an air mass tracer of dust events. These bacteria may be used as tracers of air masses during dust events, even used to distinguish the dust that has been transported over a long distance from local dust.

Furthermore, *Firmicutes* was the predominant phylum in the Gobi Desert. The proportions of this phylum reach as high as 82% in surface sand samples, but it was found in relatively small proportions that did not exceed 5% in the air samples (Fig. 14). It's clear that the bacterial community composition in the air is very different from that in the surface sand or soil.

The predominant fungal phyla were *Ascomycota* (mainly *Dothideomycetes* and *Sordariomycetes*) and *Basidiomycota* (mainly *Agaricomycetes*), and there was also much lower relative amount of *Chytridiomycota* (Fig. S5). There is no obvious pattern of the predominant fungal phyla in dust and non-dust samples. At class level, *Agaricomycetes*, *Dothideomycetes* and *Sordariomycetes* were the predominant, and dust samples contained more diverse fungal classes than non-dust samples (Fig. 16). As for Erenhot, the relative amounts of *Microbotryomycetes*, *Pucciniomycetes* and *Tremellomycetes* increased significantly compared with that of non-dust samples. While the relative amounts of *Eurotiomycetes* in dust samples of R-*DzToUb* had a remarkable boom, in the meantime, that of *Agaricomycetes* almost halved in dust samples, comparing with that of non-dust samples there. In conclusion, there are obvious difference of the fungal community composition between dust and non-dust samples, and the changing pattern may be diversiform within different locations.



## 4 Conclusion

During the DuBi-2016 campaign, bioaerosol samples were continuously collected along the transport path of Asian dust, and the effects of dust events originating in the Gobi Desert of Asia on the amounts and diversity of bioaerosols were investigated. The concentrations of DAPI-stained bacteria in the dust samples can reach two orders of magnitude greater than those observed in the non-dust samples and three orders of magnitude greater for the yellow particles (organic matter). In addition, the alpha diversity of the bacteria in the dust samples was also greater than that noted in the non-dust samples. In conclusion, both the number of bacteria and the diversity of the bacterial communities increased significantly during the dust events, as determined by means of microscopic observations made with DAPI staining and MiSeq sequencing analysis. It indicates that dust events can carry a surprising number of highly diverse microbes to downwind regions, and this transport may have potentially important impacts on local ecological conditions and climate change.

Although deserts likely play a less important role as a source of biological matter to the atmosphere than do biologically active regions, the atmospheric residence time of particles emitted from deserts is much longer than for most other source regions as a result of the combination of strong dry convection and a lack of removal by precipitation in desert regions (Burrows et al., 2009; Schulz et al., 1998). So bioaerosols in the dust particles emitted there are more likely to participate in long-distance transport and be observed in other regions. During the long-distance transport period, airborne microbes employ diverse strategies to adapt to the harsh atmospheric environment and maintain their viability. Microbial activities, including reproductive activity, may take place in the air, as were partly established by microscopic and SEM observations. Reproductive activity increases the number of microbes in the

air, which may lead to underestimation of the concentrations of microbes. Some activities may change

the physicochemical properties of other atmospheric components, such as secondary organic aerosols,

thereby changing their capacity to serve as ice nuclei (INs) or cloud condensation nuclei (CCNs), their

radiative properties and their other characteristics.

The predominant bacterial phyla found in the air samples were *Actinobacteria*, *Bacteroidetes*,

*Proteobacteria*. Many bacterial members can enter the atmosphere with the aid of wind. Some bacterial

members display no resistance to the harsh environmental stressors in the atmosphere and are

eliminated mostly; on the other hand, the remainder can be transported over long distances on the wind

and have long-term impacts on ecological environments and climate change. *Firmicutes* provides a

good example; it was found in large proportions that may even reach 82% at the surface of the Gobi

Desert. However, most of them were eliminated by environmental stressors, and only a small fraction

remained and the relative amounts in the air were less than 5%. Moreover, the relative amounts of some

bacterial members and fungal members increased markedly, together with the higher alpha diversity in

dust samples than that in non-dust samples, which contributed to a high diversity of the bacterial

community in the downwind atmosphere, potentially representing a threat to local ecological

environments.

Naturally, the dust and non-dust samples could be clearly separated from each other, due to the

differing compositions of the bacterial communities they contained. In addition, some bacteria and fungi

were only found in the dust samples. These taxa may originate in the dust source regions and can be

used for provenance tracking, particularly to distinguish dust transported over long distances from local

dust.

Bioaerosols originating from Asian desert areas have high possibility to disperse to downwind regions, such as Korea and Japan, by the prevailing westerly winds in the middle latitudes (Iwasaka et al., 2009) and are sometimes carried to the Pacific Ocean (Smith et al., 2013). Huge dust events create an atmospheric bridge over land and sea, which may contribute to the biodiversity on the earth, but the

impact of bioaerosols transported over long distance should be checked carefully.

Although the amount and diversity of bioaerosols in the air have been investigated, the viability and activity of airborne microbes, the interactions between bioaerosols and other gaseous and solid components in the air, and the effects of bioaerosols on animals and plants, the ecological environment and the climate system require in-depth study to permit a detailed understanding of bioaerosols in the

air.

**Acknowledgments**

This research was supported by the National Natural Science Foundation of China (41575017, 41375031, 41521004, 41405113 and 41505011), a China 111 project (B 13045), and the Fundamental Research Funds for the Central Universities (lzujbky-2017-k03 and lzujbky-2017-it21). This study was

also supported by Grants-in-Aid for Scientific Research (A) (No. 17H01616) and (B) (No. 26304003) from the Japan Society for the Promotion of Science (JSPS). The employees of Fasmac Co., Ltd. assisted in the MiSeq sequencing analyses.

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





**Table 1 Information of sampling sites**

| Location | Latitude (°N) and longitude (°E) | Altitude (ASL*) | Sampling height (AGL*) | Sampling period | Sampling duration |
|---|---|---|---|---|---|
| Erenhot | ER: 43.668, 111.953 | 957 m | 20 m (on a building) | 2016/3/30 - 2016/5/20 | 2 h ~24 h |
| Zhangbei | ZB: 41.156, 114.701 | 1395 m | 4 m (on a container) | 2016/3/29 - 2016/5/31 | 2 h ~16 h |
| Jinan | JN: 36.673, 117.057 | 48 m | 25 m (on a building) | 2016/3/23 - 2016/6/4 | 10 ~14 h |
| R-DzToUb | Dz: 43.557, 104.419 Ub: 47.886, 106.906 | Dz: 1489 m Ub: 1302 m | 2 m (on a car) | 2016/5/5 | 1 h |

*ASL: above sea level
*AGL: above ground level





**Fig. 1 The design of the Dust-Bioaerosol Campaign in 2016, the locations of the sampling sites and the contexts of the bioaerosol samplers (R-DzToUb is located along the road between Dalanzadgad and Ulaanbaatar).**






**Fig. 2 LiDAR observations in Zamynuud during the sampling period. D1-D7 represent dust events that occurred in Erenhot.**



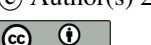

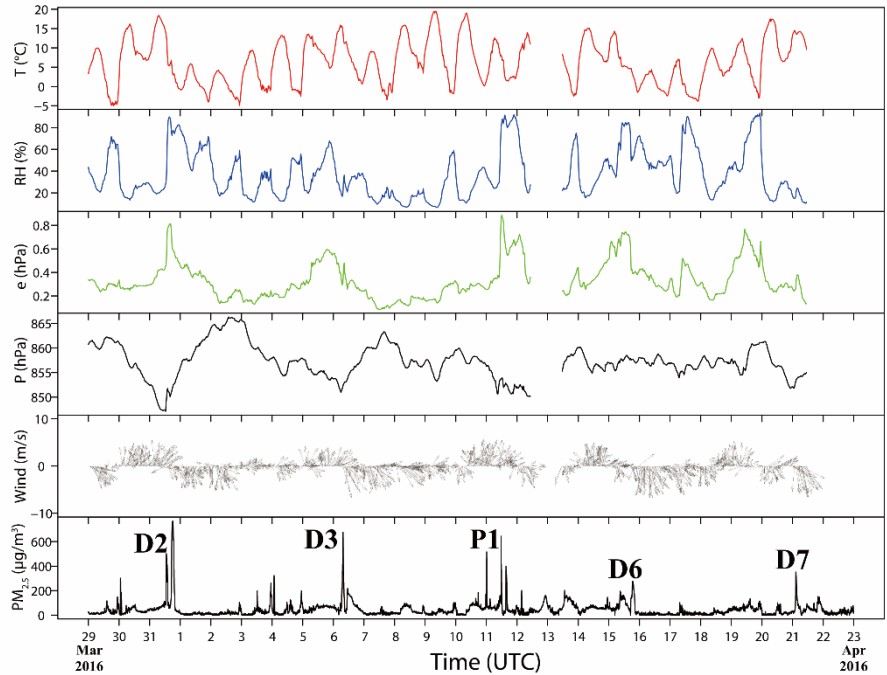

**Fig. 3 Meteorological conditions and air quality measurements during the sampling period. T, temperature; RH, relative humidity; e, water vapor pressure; P, atmospheric pressure. D1-D7 correspond to the 7 dust events that occurred in Erenhot.**





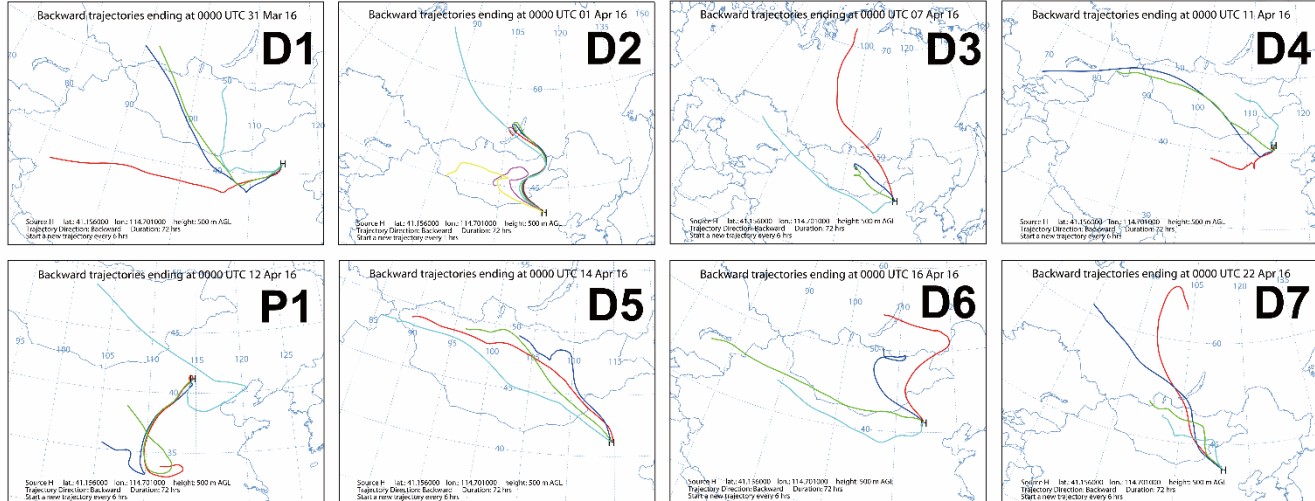

**Fig. 4 72-h back trajectories of air masses in Zhangbei calculated using the HYSPLIT model. D1-D7 represent the dust events that occurred in Erenhot.**



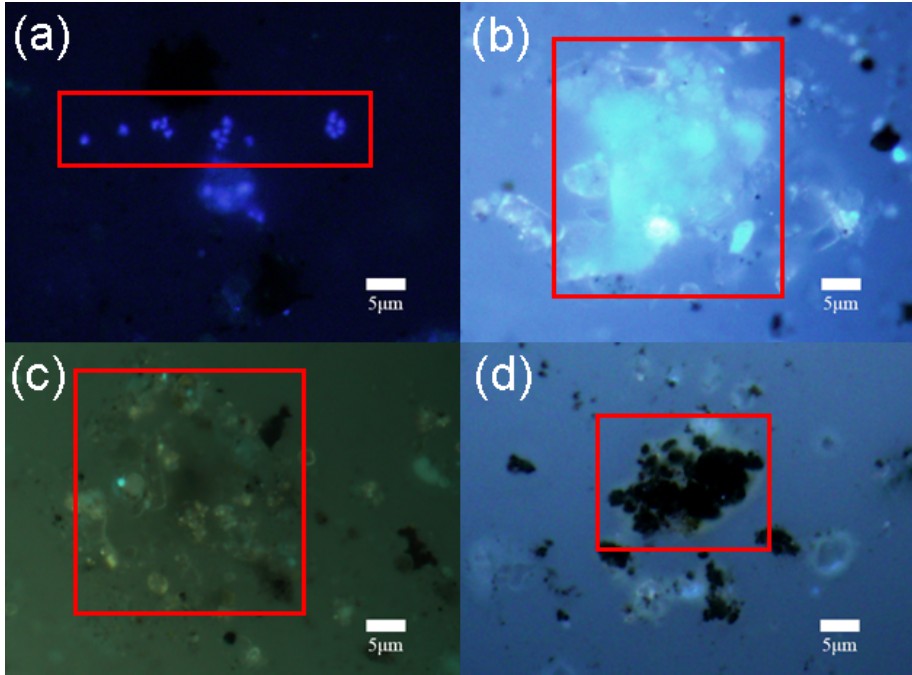

**Fig. 5 Epifluorescence micrograph of (a) DAPI-stained bacteria (with diameters < 3 μm), (b) white particles (mineral particles), (c) yellow particles (organic matter) and (d) black particles (black carbon) in air samples. All photomicrographs were taken at a magnification of × 1000.**



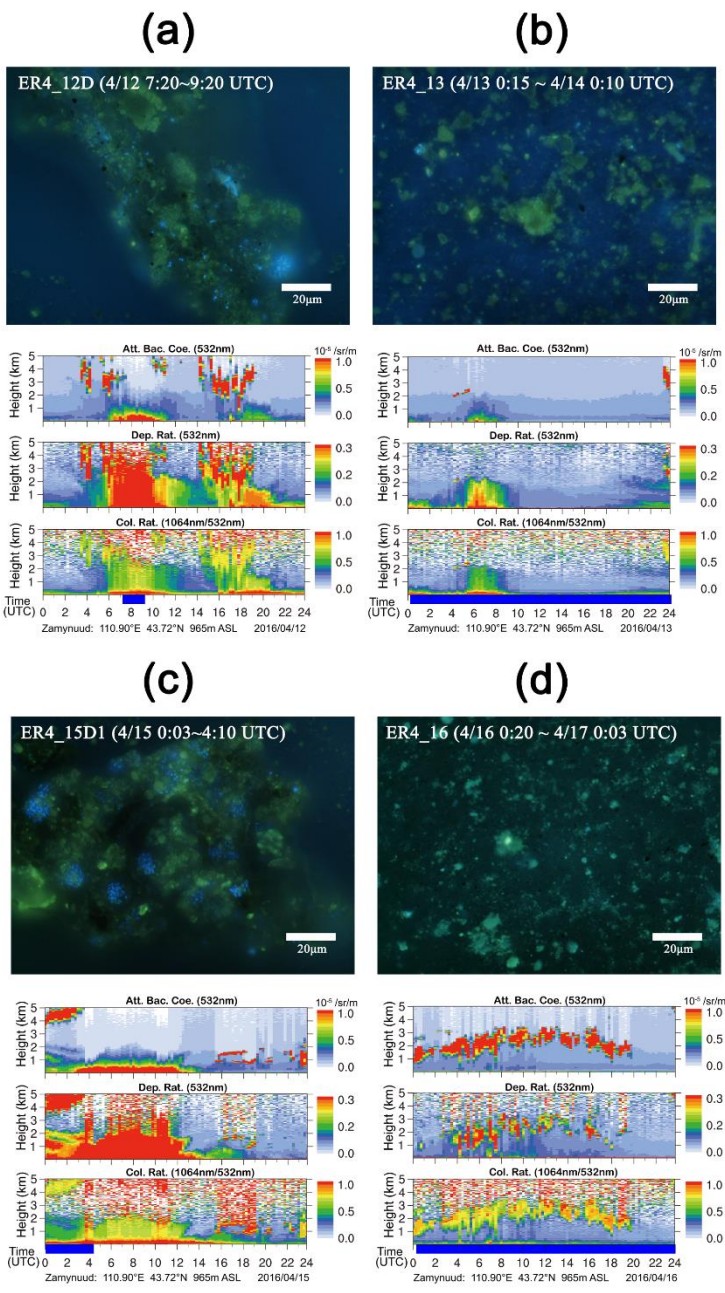

**Fig. 6 Comparisons of bioaerosols collected during a dust event and a non-dust event (a and b) and during a dust event that transported dust over a long distance and a local dust event (c and d). Blue bars represent the periods over which samples were collected.**





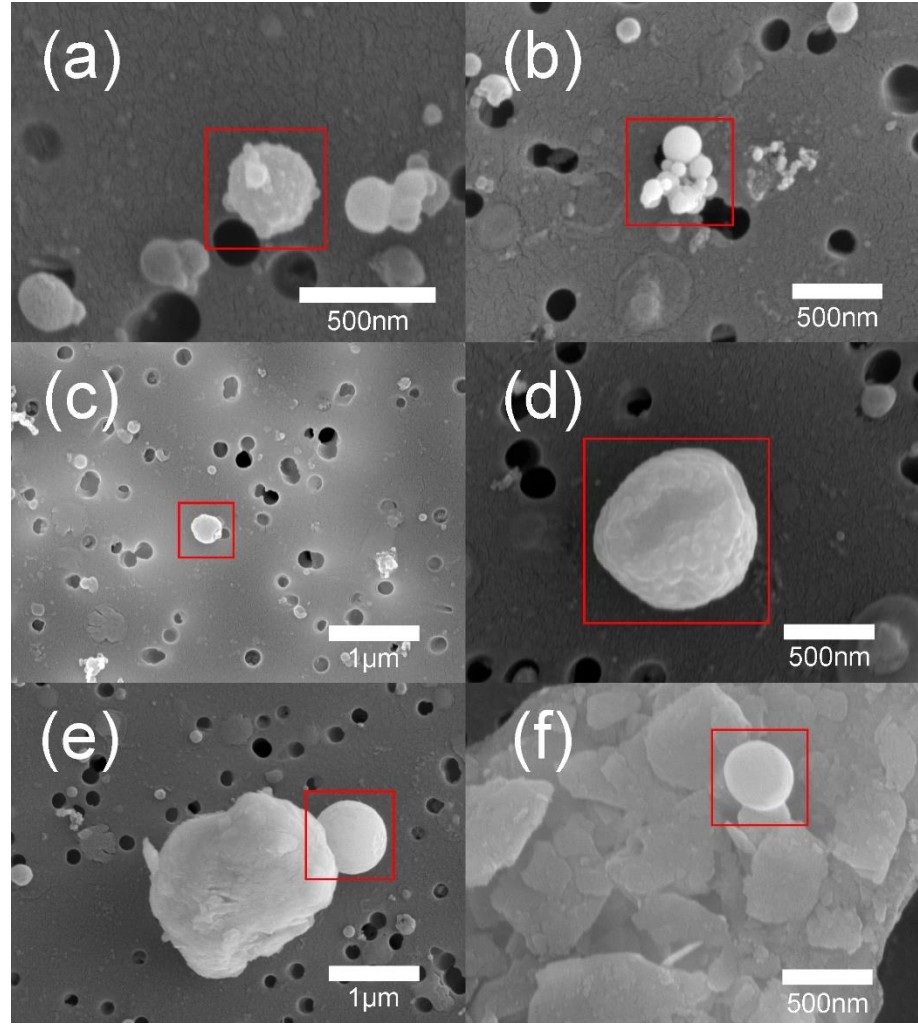

**Fig. 7 State of bioaerosols under SEM. All of the pictures depict non-dust samples collected in Erenhot (panel a shows ER4_11N, whereas the others are from ER4_9).**



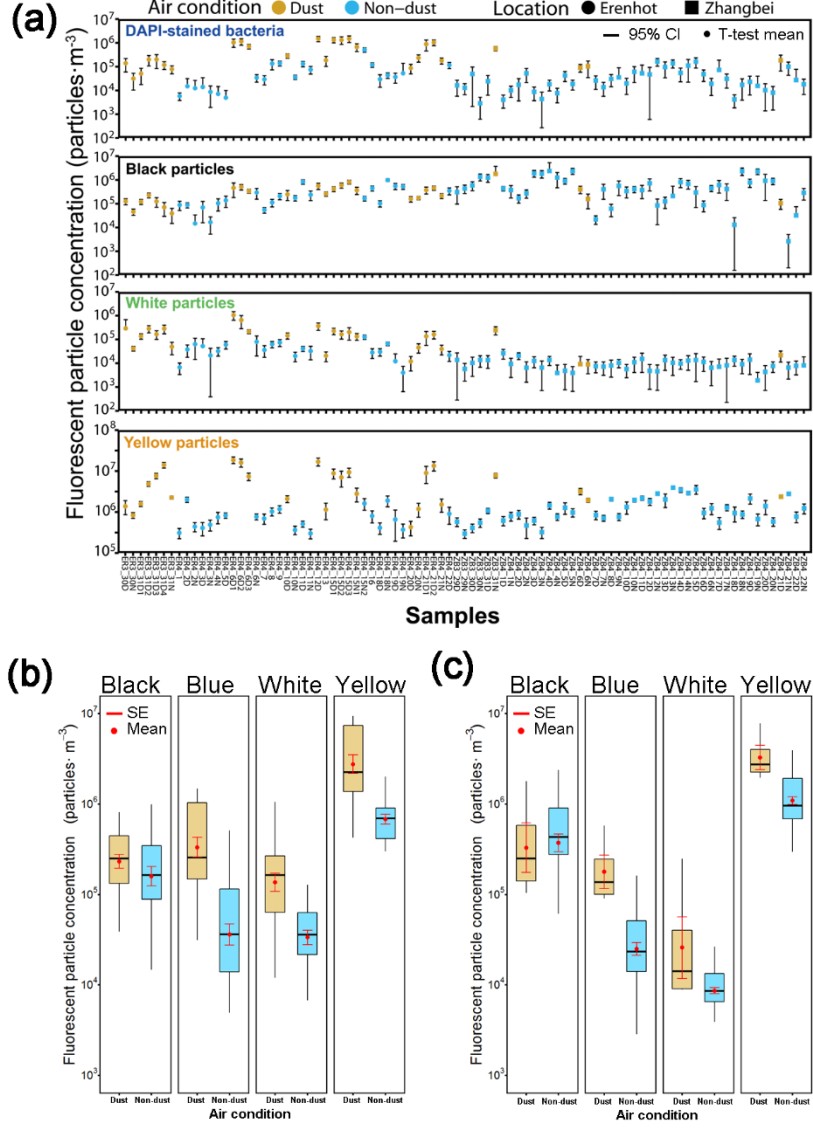

**Fig. 8 Changes in the concentrations of fluorescent particles in the samples (a) and a comparison of the concentrations of fluorescent particles in the samples collected during dust events and non-dust events in Erenhot (b) and Zhangbei (c) (Black: black particles, Blue: DAPI-stained bacteria, White: white particles, Yellow: yellow particles, CI: confidence interval, SE: standard error).**



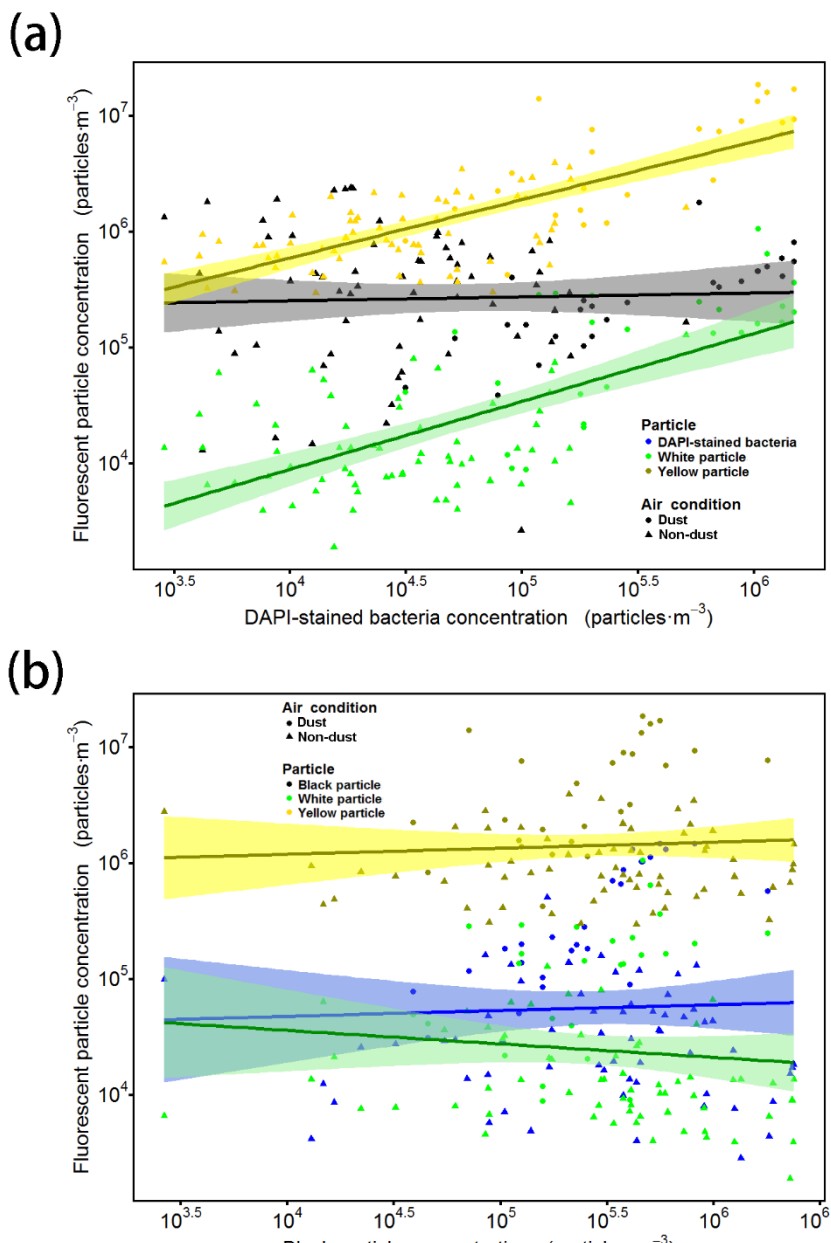

**Fig. 9 Changes in the concentrations of the other three kinds of fluorescent particles with the concentrations of DAPI-stained bacteria (a) and the concentrations of black particles (b) in the dust and non-dust samples.**





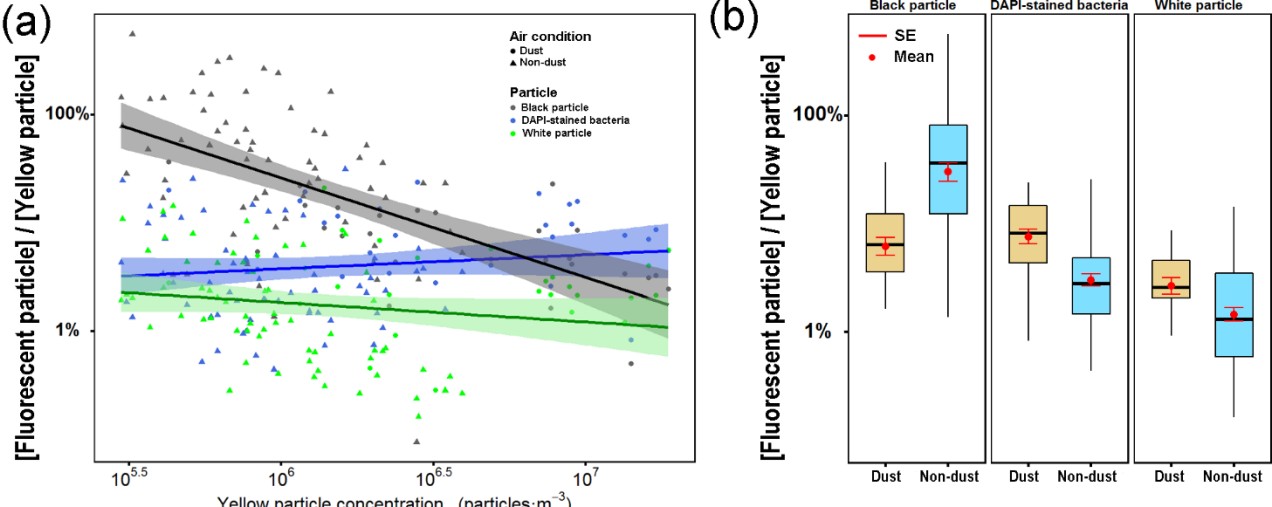

**Fig. 10 Ratios of the concentrations of three kinds of fluorescent particles to that of the yellow particles (a) and a comparison of these ratios in the dust and non-dust samples (b) (SE: standard error).**



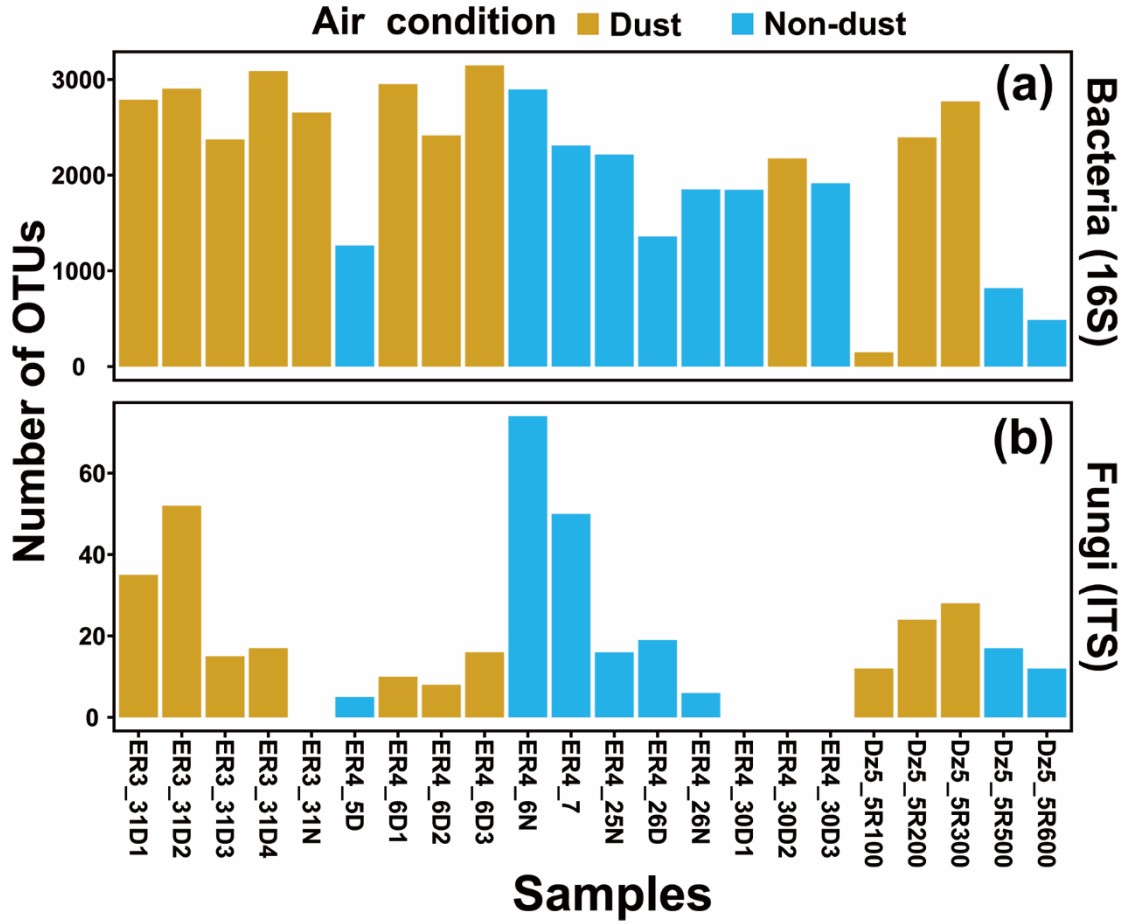

**Fig. 11 Number of OTUs of bacteria (a) and fungi (b).**





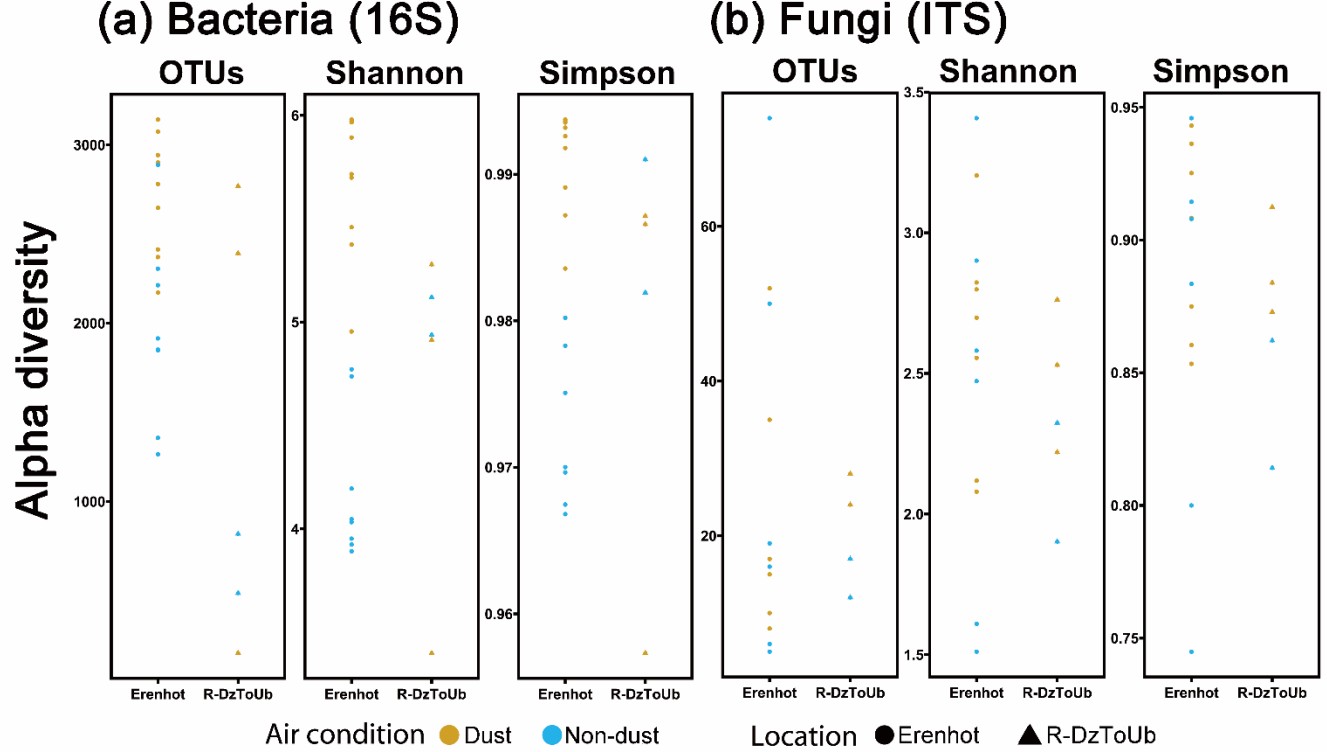

**Fig. 12 Alpha diversity of bacteria (a) and fungi (b) (OTUs: Number of OTUs, Shannon: Shannon index, Simpson: Simpson index).**



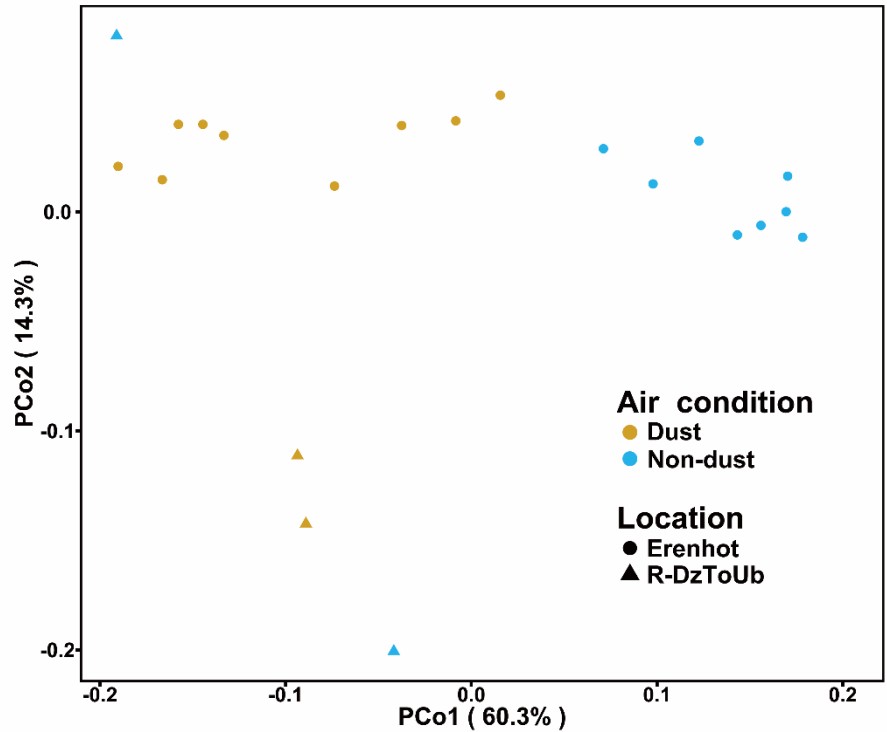

**Fig. 13 Principal coordinates analysis of bacterial 16S rDNA sequencing data obtained from 22 samples (PCo: principal coordinate).**




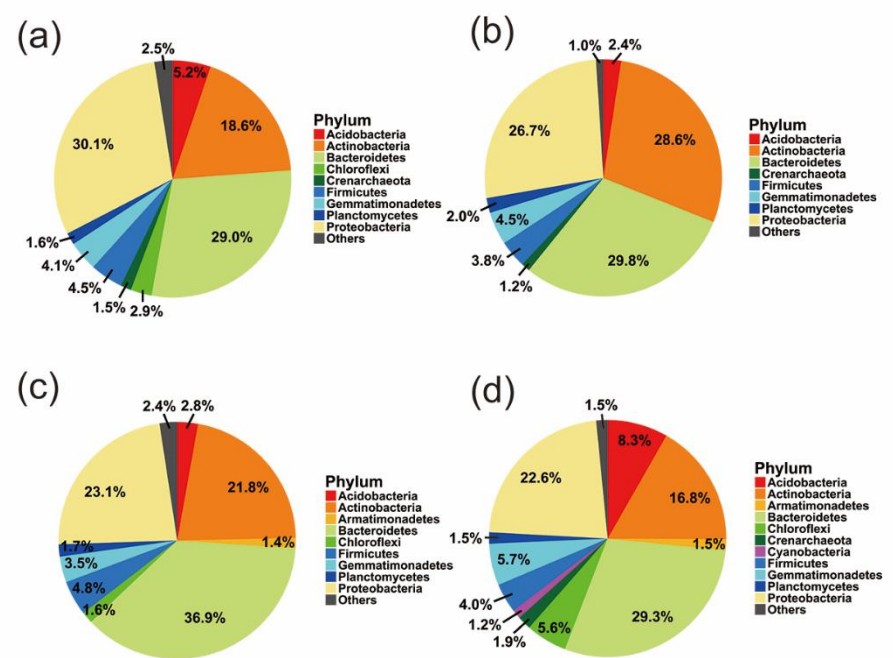

**Fig. 14 Variations of the bacterial community composition at the phylum level (a) Dust samples of Erenhot, (b) Non-dust samples of Erenhot, (c) Dust samples of R-DzToUb and (d) Non-dust samples of R-DzToUb.**

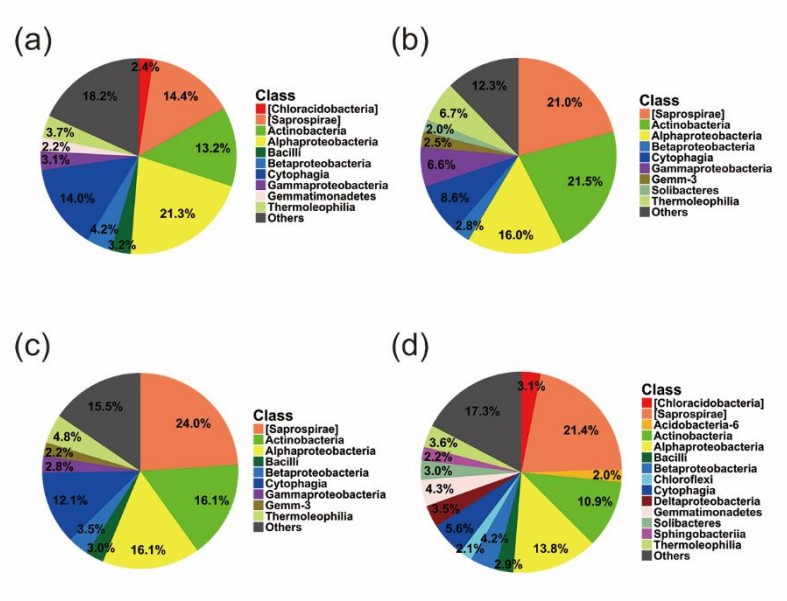





**Fig. 15 Variations of the bacterial community composition at the class level (a) Dust samples of Erenhot, (b) Non-dust samples of Erenhot, (c) Dust samples of R-DzToUb and (d) Non-dust samples of R-DzToUb.**

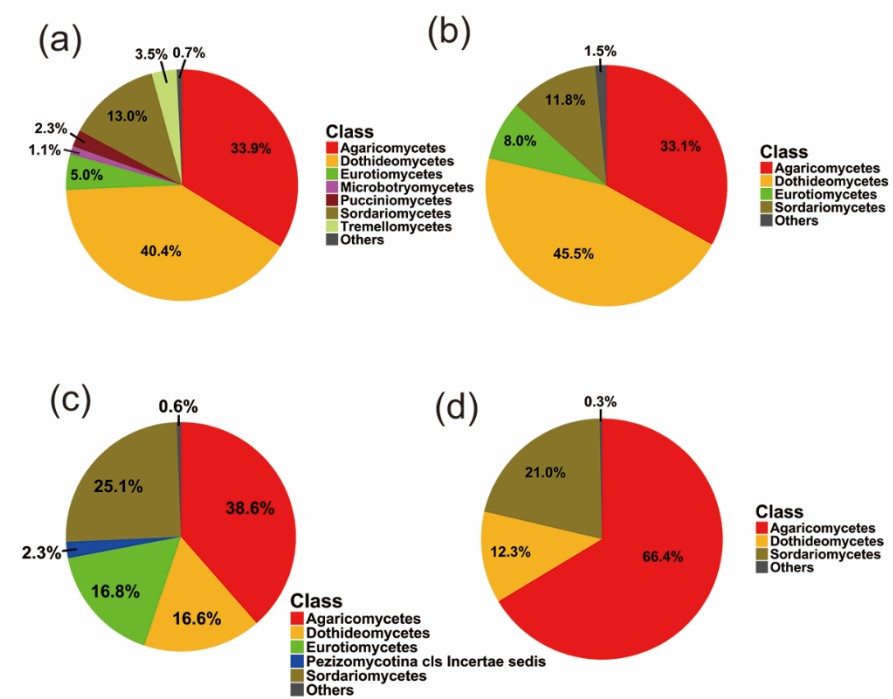

5    **Fig. 16 Variations of the fungal community composition at the class level (a) Dust samples of Erenhot, (b) Non-dust samples of Erenhot, (c) Dust samples of R-DzToUb and (d) Non-dust samples of R-DzToUb.**