# Peer review of "Characterization of atmospheric bioaerosols along the transport pathway of Asian dust during the Dust-Bioaerosol 2016 Campaign"

_Atmospheric Chemistry and Physics, 2017_

## Short Comment (SC1) · 23 Dec 2017

A comment on the lidar data.

The lidar data in Zamynuud, Mongolia is taken from the AD-Net. We request to follow the data policy of AD-Net written in the AD-Net web page at http://www-lidar.nies.go.jp/AD-Net/. Nishizawa et al., 2016 in Reference is not suitable for the observations in Mongolia. Shimizu et al., Optical Engineering 56(3), 031219, 2016. doi:10.1117/1.OE.56.3.031219 would be fine. "LiDAR in the caption of Fig. 2 should be "Lidar." The term LiDAR is used only in the lidar mapping community.

---

## Author Comment (AC1) · 27 Dec 2017

We are very grateful for the SC1's critical comments and suggestions, which have helped us improve the paper quality. We have add 'Nishizawa et al., 2016' in References, and add 'Lidar data were provided by courtesy of AD-Net (http://www-lidar.nies.go.jp/AD-Net).' in Acknowledgement. 'LiDAR' in the caption of Fig. 2 has also been changed to 'Lidar'. In addition, we have done other modifications as required. We have uploaded the modified manuscript in the form of the attached supplement and figures.

**Fig. 1.** First example of the mixed-type aerosols

8000

[Figure]

**10 µm**

**Fig. 2.** Second example of the mixed-type aerosols

---

## Short Comment (SC2) · 30 Dec 2017

In the Author Comment "AC1: 'Reply to SC1', Tang Kai, 27 Dec 2017", "We have add 'Nishizawa et al., 2016' in References, and add 'Lidar data were provided by courtesy of AD-Net (http://wwwlidar.nies.go.jp/AD-Net).' in Acknowledgement." should be "We have added 'Shimizu et al., 2016' in References, and added 'Lidar data were provided by courtesy of AD-Net (http://wwwlidar.nies.go.jp/AD-Net).' in Acknowledgement.".

Very sorry about this mistake, and it may be misunderstood by others. Actually, I have modified the original manuscript according to Dr. Sugimoto's suggestions, and the modified manuscript was attached as the Supplement, which can be seen by the

editor.

In addition, I am very sorry about that I neglected the following data policy of AD-Net, "When the user submits a paper to some journal or makes oral presentation utilizing data from AD-Net, the user has to obtain permission from data manager (currently A. Shimizu) for the usage." Considering the importance of the lidar data from AD-Net in this publication, we have invited Dr. Shimizu as one of co-authors. Many thanks for the outstanding work of all the members who contribute to AD-Net, and thank Dr. Sugimoto for his kindness.

---

## Referee Comment (RC1) · K.C. Lee (Referee) · 25 Jan 2018

Tang et al., have conducted a large scale and comprehensive campaign with the aim to explain key questions in the nature of long distance aerosol transport. The manuscript documents two major aspects of the aerosol samples: microscopy and molecular biology, through several sampling sites and spanning several dust events. The paper provided valuable findings relating an important process (Asian dust events) which may be applicable to many other similar systems worldwide.

Pg.1, Ln. 22 - I do not understand what "charge capacity" means throughout this paper. I can not find relevant results regarding "charge capacity" and microscopy. Did you

mean something like fluorescence intensity, fluorescence concentration (as indicated when referring to some of the figures), or particle counts?

Pg.5, Ln.7 - First mention of MiSeq should have company information (Illumina, CA, USA)

Pg.7, Ln.7 - in-text citation style should be "...previously described by Maki et al., (2014)." This should also be changed in other parts of the manuscript.

Pg.8, Ln.5 - Suggest changing to "phenol chloroform extraction/ ethanol precipitation" for clarity.

Pg.8, Ln.6 - in-text citation style should be "Maki et al., (2017)"

Pg.8, Ln.10 - Should include the hypervariable region(s) targeted (and the primer used) in the first step of PCR amplification.

Pg.8, Ln.12 - This BioProject is not publicly available, it needs to be released.

Pg.9, Ln.16 - Consistency in period symbol: (Att. Bac. Coe.) (Dep. Rat.) (Col. Rat.)

Pg.14, Ln.4 - This seems to be a major observation/trend, why do you think this is the case?

Pg.14, Ln.6 - For reproducibility, please consider uploading the OTU sequences as supplemental files in FASTA format.

Pg.15, Ln.21 - Avoid contraction, use "It is"

Pg.17, Ln.9 - Avoid contraction, use "It is"

---

## Referee Comment (RC2) · Anonymous Referee #2 · 9 Mar 2018

Bioaerosols are a class of atmospheric particles, which are more likely to participate in long-distance transport and be observed in other regions. This manuscript investigated the effects of dust events originating in the Gobi Desert of Asia on the amount and diversity of bioaerosols. In this study, sufficient and comprehensive experimental data was presented to reveal that the number of bacteria and the diversity of the bacterial communities showed remarkable increases during the dust events. Microscopic observations made with DAPI staining and MiSeq sequencing analysis were used to determine the results. In general, this manuscript was well-organized and the main conclusions will help improve the current understanding of bioaerosol dynamics along the transport pathway of Asian dust in China. This manuscript should be published

in ACP after a little more discussion and analysis to clarify the details behind the presented results. Specific comments: 1. Page 5 line 3. The result of Jinan samples should be compared with the result from another bioaerosol campaign (AAQR 18, 1-14, 2018). 2. Page 9, line 20 – page 10 line 4. The concentrations of PM2.5 increased significantly in Zhangbei during D2, D3 and D7. It seems that Zhangbei was seriously affected by the dust events in Erenhot. But the next part of the paper said that Zhangbei was slightly affected by the dust events based on D6. The authors need to explain it. 3. Page10 line 19. The name of the sample should be ER4_12D. Please be sure that all the samples' names are correct. 4. Page14 line6. The analyzed 22 samples were collected in Erenhot or Zhangbei? 5. In Fig. 4. Please clarify the meaning of different colors of the air masses. 6. In Fig.9 and in Fig. 10(a). How the authors get the fitting curves and fitting areas. Please clarify it.

―――――――――――――――――――――――

---

## Referee Comment (RC3) · Anonymous Referee #3 · 25 Mar 2018

This paper examines atmospheric bioaerosols at three sites downwind of the Gobi Desert in the Dust-Bioaerosol 2016. The authors found that the number of bacteria and the diversity of the bacterial communities increased significantly during the dust events by microscopic observations made with DAPI staining and MiSeq sequencing analysis. In general, this is a well-written paper that presents interesting data. It will be of interest to readers of this Journal, particularly researchers in the field.

Page 2, line 6: "proteobacteria" should be capitalized.

Page 8: The description of the methods of MiSeq sequencing should be limited. It would help readers if the authors gave a more detailed explanation.

[Figure]

Page 10, line 18 to Page 11, lines 8, Fig. 6, and Table S1: The sample names contain a number of errors. Please check all sample names, including sampling information, and revise them accordingly.

Page 14, lines 11 and 14: "orders (and class level candidate taxa)" should be "orders (and order-level candidate taxa)".

Fig. 9: The authors should check the symbols in Fig. 9. "DAPI-stained bacteria" should be "Black particle" in Fig. 9 (a). In contrast, "Black particle" should be "DAPI-stained bacteria" in Fig. 9 (b).
* * *

---

## Author Comment (AC2) · 5 Apr 2018

**Response to Referee #1's Comments**

**General Comments**

Tang et al., have conducted a large scale and comprehensive campaign with the aim to explain key questions in the nature of long distance aerosol transport. The manuscript documents two major aspects of the aerosol samples: microscopy and molecular biology, through several sampling sites and spanning several dust events. The paper provided valuable findings relating an important process (Asian dust events) which may be applicable to many other similar systems worldwide.

**Response: We thank the reviewer for his useful comments and suggestions. Those comments and suggestions helped us a lot to improve the quality of this paper. The authors have taken the comments from reviewers seriously and addressed all comments in current revision. Below are our point-by-point responses to those comments.**

**Specific Comments**

Pg.1, Ln. 22 - I do not understand what "charge capacity" means throughout this paper. I can not find relevant results regarding "charge capacity" and microscopy. Did you mean something like fluorescence intensity, fluorescence concentration (as indicated when referring to some of the figures), or particle counts?

**Response:** The charge capacity means the ratios of the concentrations of two kinds of fluorescent particles. For example, the charge capacity of yellow fluorescent particles associated with the DAPI-stained bacteria means the ratios of the concentrations of DAPI-stained bacteria to those of yellow fluorescent particles. It can be referred to Figure 10 (a) and (b). In Figure 10, "[Fluorescent particle]/[Yellow particle]" means ratios of the concentrations of three kinds of fluorescent particles to those ('that' has been corrected) of the yellow particles. In order to avoid the reader's misunderstanding, '**the charge capacity**' has been replaced by '**the concentration ratios**' throughout the manuscript.

**Original Text Pg.1 Ln.22**: Moreover, the charge capacity of yellow fluorescent particles associated with the DAPI-stained bacteria increased from 5.1% ± 6.3% (non-dust samples) to 9.8% ± 6.3% (dust samples).

**Amended Text Pg.1 Ln.22:** Moreover, the concentration ratios of DAPI-stained bacteria to yellow fluorescent particles increased from 5.1% ± 6.3% (non-dust samples) to 9.8% ± 6.3% (dust samples).

Pg.5, Ln.7 - First mention of MiSeq should have company information (Illumina, CA, USA)

**Response:** By following the reviewer's suggestion, we have added '(Illumina, CA, USA)' when MiSeq is mentioned at the first time.

Pg.7, Ln.7 - in-text citation style should be "...previously described by Maki et al., (2014)." This should also be changed in other parts of the manuscript.

**Response:** By following the reviewer's suggestion, it has been corrected throughout the manuscript.

Pg.8, Ln.5 - Suggest changing to "phenol chloroform extraction/ethanol precipitation" for clarity.

**Response:** Thank the reviewer for helpful suggestions, we have modified it as you suggested.

Pg.8, Ln.6 - in-text citation style should be "Maki et al., (2017)"

**Response:** We thank the reviewer for the helpful suggestion, and have corrected it.

Pg.8, Ln.10 - Should include the hypervariable region(s) targeted (and the primer used) in the first step of PCR amplification.

**Response:** "During the first-step PCR amplification, fragments of 16S rRNA gene (which covered the variable region V4) were amplified from the extracted gDNA using the universal bacterial primers 515F (5'-Seq A-TGTGCCAGCMGCCGCGGTAA-3')

and 806R (5'-Seq B-GGACTACHVGGGTWTCTAAT-3') (Caporaso et al., 2011), where Seq A and Seq B represent the nucleotide sequences bounded by the primer sets of second-step PCR. Detailed process has been described by Maki et al. (2017)." has been added in '2.4 DNA extraction, sequencing and phylogenetic analysis'.

Pg.8, Ln.12 - This BioProject is not publicly available, it needs to be released.

**Response:** By following the reviewer's suggestion, the BioProject PRJNA413598 has been released on 2018-03-19.

Pg.9, Ln.16 - Consistency in period symbol: (Att. Bac. Coe.) (Dep. Rat.) (Col. Rat.)

**Response:** We thank the reviewer for helpful suggestion, the missing dots have been added throughout the manuscript.

Pg.14, Ln.4 - This seems to be a major observation/trend, why do you think this is the case?

**Response:** The major trend is "greater numbers of bacteria can be contained in a unit of yellow particles during dust events, whereas the black particles displayed the opposite behavior". Yellow particles (organic matter) can serve as nutrient sources for microbes, and favor their survival and long-distance transport. In addition, some dead microbes also emit yellow fluorescence. Therefore, it's reasonable that greater numbers of bacteria can be contained in a unit of yellow particles during dust events. On the contrary, more black particles (black carbon) was contained in a unit of yellow particles during non-dust events compared with dust events. Meanwhile, it is speculated that anthropogenic black carbon emission has a significant increase during non-dust periods comparing with that in dust events. It's worth noting that the mixing of the dust and black carbon during the long-distance transport (Fig. S4). Some researches show that the comparison of dust aerosols and anthropogenic pollutants (such as black carbon) shows a clear distinction of optical and radiative characteristics (Huang et al., 2011; Pu et al., 2015; Wang et al., 2010, 2013, 2014, 2015, 2017). Hence the further assessment of the radiative effects of the mixed-type aerosols is warranted.

**Based on the above explanation, the manuscript has been revised as follows:**

[revised manuscript text omitted]

increased significantly during non-dust periods comparing to dust event periods.

[Figure]

Fig. S4 Epifluorescence micrograph of mixed-type aerosols, (a) from the sample ER4_15D2, (b) and (d) from the sample ER4_11N, (c) from the sample ER4_15N1.

Pg.14, Ln.6 - For reproducibility, please consider uploading the OTU sequences as supplemental files in FASTA format.

**Response:** By following the reviewer's suggestion, all 16S OTU sequence data has been uploaded in FASTA format.

Pg.15, Ln.21 - Avoid contraction, use "It is"

**Response:** By following the reviewer's suggestion, we have corrected.

Pg.17, Ln.9 - Avoid contraction, use "It is"

**Response:** We thank the reviewer for helpful suggestions again, and have corrected it.

---

## Author Comment (AC3) · 5 Apr 2018

**Response to Referee #2's Comments**

**General Comments**

Bioaerosols are a class of atmospheric particles, which are more likely to participate in long-distance transport and be observed in other regions. This manuscript investigated the effects of dust events originating in the Gobi Desert of Asia on the amount and diversity of bioaerosols. In this study, sufficient and comprehensive experimental data was presented to reveal that the number of bacteria and the diversity of the bacterial communities showed remarkable increases during the dust events. Microscopic observations made with DAPI staining and MiSeq sequencing analysis were used to determine the results. In general, this manuscript was well-organized and the main conclusions will help improve the current understanding of bioaerosol dynamics along the transport pathway of Asian dust in China. This manuscript should be published in ACP after a little more discussion and analysis to clarify the details behind the presented results.

**Response: We would like to thank the reviewer for his positive comments and suggestions. Those comments and suggestions helped us a lot to improve the quality of this paper. The authors have taken the comments from reviewers seriously and addressed all comments in current revision. Below are our point-by-point responses to those comments.**

**Specific Comments**

Page 5 line 3. The result of Jinan samples should be compared with the result from another bioaerosol campaign (AAQR 18, 1-14, 2018).

**Response:** We thank the reviewer for helpful suggestions, it is of great importance to explore the difference of the bacterial communities between air samples (or dust samples) and cloud samples (Zhu et al., 2018). In this manuscript, 22 samples from Erenhot (17 samples) and Mongolia (5 samples) were analysed by MiSeq sequencing, while Jinan samples were not yet analysed. In spite of this, the comparison between

these 22 samples and cloud samples should be inspirational. In the cloud samples of Mt. Tai (Zhu et al., 2018), the dominant bacterial phylum was *Firmicutes*, whose averaged relative abundance was 80.5%, and *Proteobacteria*, *Bacteroidetes*, *Actinobacteria*, and *Fusobacteria* were the following. While *Bacteroidetes* and *Proteobacteria* were the dominant in these 22 air samples, followed by *Actinobacteria* (Fig. 14). The relative abundance of *Firmicutes* did not exceed 5%, and the phylum *Fusobacteria* was not found in these 22 air samples (Fig. S4).

**Original Text Pg.16 Ln.15-19 was deleted:** At the class level, *Chloracidobacteria* and *Gemmatimonadetes* in dust samples of Erenhot and non-dust samples of R-DzToUb have higher relative amounts compared with non-dust samples of Erenhot and dust samples of R-DzToUb (Fig. 15). *Cytophagia* in the phylum *Bacteroidetes* shows similar phenomenon. Further, *Bacilli* in non-dust samples of Erenhot shows very low amounts down to the detection limit, whereas its relative amounts in other samples keep stable.

**Original Text Pg.17 Ln.7:** Furthermore, *Firmicutes* was the predominant phylum in the Gobi Desert. The proportions of this phylum reach as high as 82% in surface sand samples, but it was found in relatively small proportions that did not exceed 5% in the air samples (Fig. 14). It's clear that the bacterial community composition in the air is very different from that in the surface sand or soil.

**Amended Text Pg.17 Ln.7:** The relative abundance of *Firmicutes* increased slightly in dust samples compared with non-dust samples (Fig. 14). *Firmicutes* was the predominant phylum of surface sand samples in the Gobi Desert of Asia, but not in the Taklamaken Desert (An et al., 2013). The relative abundance of *Firmicutes* could reach as high as 82% in the surface sand samples from the Gobi Desert of Asia (44.3°N, 110.1°E) (An et al., 2013), but it was found in relatively small proportions that did not exceed 5% in all the air samples (Fig. 14). Maki et al. (2016) found that the relative

abundance of *Firmicutes* in air samples from the Gobi Desert of Asia (44.2304°N, 105.1700°E) varied greatly, from 15.7 to 40.5% in non-dust samples, and no more than 12% in dust samples. The sequences of *Firmicutes* mainly belonged to the classes *Bacilli* and *Clostridia* in air samples from Tsogt-Ovoo, Mongolia (Maki et al., 2016). While *Bacilli*, *Clostridia* and *Erysipelotrichi* in the phylum *Firmicutes* were found in the air samples from Erenhot (Fig. S5). The averaged relative abundance of *Bacilli* in dust samples from Erenhot was 3.2%, while it is much lower in non-dust samples (Fig. 15). It is worth mentioning that Zhu et al. (2018) found that the averaged relative abundance of *Firmicutes* in cloud samples at Mt. Tai of China was 80.5%. As for the Taklimakan Desert, Puspitasari et al. (2015) analyzed the bacterial diversity in sand dunes and dust aerosol, and the relative abundance of *Firmicutes* in dust aerosol samples was higher than that in surface sand samples, which shows a different pattern comparing to the Gobi Desert of Asia. In conclusion, the bacterial community compositions in the air are different from that in the surface sand or soil, and differ by location and transmitting vector.

Page 9, line 20 – page 10 line 4. The concentrations of PM2.5 increased significantly in Zhangbei during D2, D3 and D7. It seems that Zhangbei was seriously affected by the dust events in Erenhot. But the next part of the paper said that Zhangbei was slightly affected by the dust events based on D6. The authors need to explain it.

**Response:** During the dust event 'D6', the PM2.5 mass concentrations showed a slight increase, not as heavy as D2, D3 and D7. In addition, the barometric pressure of D6 was relatively stable, but that of D2, D3 and D7 were on the contrary, showing a significant decrease. Thereby, Zhangbei was slightly affected by the dust events during the dust event 'D6'.

**Original Text Pg.10 Ln.2:** A slight increase in $PM_{2.5}$ mass concentrations was observed during event D6, accompanied by a strong north wind, indicating that Zhangbei was slightly affected by the dust events that occur in Erenhot.

**Amended Text Pg.10 Ln.2:** A slight increase in $PM_{2.5}$ mass concentrations was observed during the event D6, accompanied by a strong north wind and the relatively

stable atmospheric pressure, indicating that Zhangbei was slightly affected by the dust events that occur in Erenhot at that time.

Page10 line 19. The name of the sample should be ER4_12D. Please be sure that all the samples' names are correct.

**Response:** By following the reviewer's suggestion, we have corrected it throughout the manuscript.

Page14 line6. The analyzed 22 samples were collected in Erenhot or Zhangbei?

**Response:** The analyzed 22 samples were collected in Erenhot and on the road between Dalanzadgad and Ulaanbaatar (R-DzToUb). The names of the 22 samples can be referred to x-coordinate label of Fig. 11. And detailed sample information can be referred to Table S1 and Table S3.

In Fig. 4. Please clarify the meaning of different colors of the air masses.

**Response:** The different colors of the trajectories mean the different backward trajectory ending time. The trajectory with red color is the backward trajectory which has the latest ending time. Blue is the next, then green, cyan, purple, yellow.

In Fig.9 and in Fig. 10(a). How the authors get the fitting curves and fitting areas. Please clarify it.

**Response:** The fitting curves and fitting areas were plotted by R software, and the function stat_smooth (method=lm, level=0.95) in the package 'ggplot2' was used. The fitting curves were calculated by the linear regression method, and the independent variable was sorted before regression. The fitting areas were calculated based on the confidence interval of 95%.

---

## Author Comment (AC4) · 5 Apr 2018

**Response to Referee #3's Comments**

**General Comments**

This paper examines atmospheric bioaerosols at three sites downwind of the Gobi Desert in the Dust-Bioaerosol 2016. The authors found that the number of bacteria and the diversity of the bacterial communities increased significantly during the dust events by microscopic observations made with DAPI staining and MiSeq sequencing analysis. In general, this is a well-written paper that presents interesting data. It will be of interest to readers of this Journal, particularly researchers in the field.

**Response: We sincerely thank the reviewer for his suggestions. Those suggestions helped to improve the quality of this paper. The authors have taken the comments from reviewer seriously and addressed all comments in current revision. Below are our point-by-point responses to those comments.**

**Specific Comments**

Page 2, line 6: "proteobacteria" should be capitalized.

**Response:** By following the reviewer's suggestion, we have corrected it.

Page 8: The description of the methods of MiSeq sequencing should be limited. It would help readers if the authors gave a more detailed explanation.

**Response:** We thank the reviewer for the helpful suggestion, and have revised section 2.4 as follow.

**Original Text Pg.8 Ln.4:** The genomic DNA (gDNA) was extracted from the atmospheric samples from Erenhot and Mongolia using the PC extraction/alcohol precipitation method. Two-step PCR amplification and product purification were then carried out according to the method of Maki (Maki et al., 2017). Two-step PCR has several advantages, such as increased reproducibility and the recovery of greater levels of genetic diversity during amplicon sequencing (Park et al., 2016). An Illumina MiSeq sequencing system (Illumina, CA, USA) and a MiSeq Reagent Kit V2 (Illumina, CA,

USA) were used to perform the sequencing, and an average read length of 270 bp was obtained. All the data obtained from MiSeq sequencing have been deposited in the DDBJ/EMBL/GenBank database, and the accession number of the submission is PRJNA413598.

**Amended Text Pg.8 Ln.4:** The genomic DNA (gDNA) was extracted from the atmospheric samples from Erenhot and Mongolia using the phenol chloroform extraction/ethanol precipitation method (Maki et al., 2017). Two-step PCR amplification and product purification were then carried out according to the method of Maki et al. (2017). Two-step PCR has several advantages, such as increased reproducibility and the recovery of greater levels of genetic diversity during amplicon sequencing (Park et al., 2016). During the first-step PCR amplification, fragments of 16S rRNA (which covered the variable region V4) were amplified from the extracted gDNA using the universal bacterial primers 515F (5'-Seq A-TGTGCCAGCMGCCGCGGTAA-3') and 806R (5'-Seq B-GGACTACHVGGGTWTCTAAT-3') (Caporaso et al., 2011), where Seq A and Seq B represent the nucleotide sequences bounded by the primer sets of second-step PCR. Detail process has been described by Maki et al. (2017). An Illumina MiSeq sequencing system (Illumina, CA, USA) and a MiSeq Reagent Kit V2 (Illumina, CA, USA) were used to perform the sequencing, and an average read length of 270 bp was obtained. All the data obtained from MiSeq sequencing have been deposited in the DDBJ/EMBL/GenBank database, and the accession number of the submission is PRJNA413598.

Page 10, line 18 to Page 11, lines 8, Fig. 6, and Table S1: The sample names contain a number of errors. Please check all sample names, including sampling information, and revise them accordingly.

**Response:** We thank the reviewer for the helpful suggestion, and have corrected all sample names throughout the manuscript and checked the sampling information in the supplement. In Pg.10 Ln.19, the sample name 'ER4_12' has been corrected to 'ER4_12D'.

Page 14, lines 11 and 14: "orders (and class level candidate taxa)" should be "orders (and order-level candidate taxa)".

**Response:** By following the reviewer's suggestion, we have corrected it.

Fig. 9: The authors should check the symbols in Fig. 9. "DAPI-stained bacteria" should be "Black particle" in Fig. 9 (a). In contrast, "Black particle" should be "DAPI-stained bacteria" in Fig. 9 (b).

**Response:** We thank the reviewer for the helpful suggestion, and have corrected the symbols in Fig. 9.

---

## Author Comment (AC6) · 5 Apr 2018

**Table S1 Sampling information of Erenhot.**

[revised manuscript text omitted]